# Addition of Bee Products in Diverse Food Sources: Functional and Physicochemical Properties

Gloria Isabel Camacho-Bernal [1], Nelly del Socorro Cruz-Cansino [1,*], Esther Ramírez-Moreno [1], Luis Delgado-Olivares [1], Quinatzin Yadira Zafra-Rojas [1], Araceli Castañeda-Ovando [2] and Ángela Suárez-Jacobo [3]

[1] Instituto de Ciencias de la Salud, Universidad Autónoma del Estado de Hidalgo, Circuito Ex Hacienda La Concepción S/N, Carretera Pachuca-Actopan, San Agustín Tlaxiaca 42160, Hidalgo, Mexico; gloria_camacho11174@uaeh.edu.mx (G.I.C.-B.); esther_ramirez@uaeh.edu.mx (E.R.-M.); ldelgado@uaeh.edu.mx (L.D.-O.); quinatzin_zafra@uaeh.edu.mx (Q.Y.Z.-R.)

[2] Instituto de Ciencias Básicas e Ingeniería, Universidad Autónoma del Estado de Hidalgo, Carretera Pachuca-Tulancingo Km. 4.5, Mineral de la Reforma 42184, Hidalgo, Mexico; ovandoa@uaeh.edu.mx

[3] Centro de Investigación y Asistencia en Tecnología y Diseño del Estado de Jalisco A.C., Biotecnología Industrial, Camino Arenero 1227, el Bajío del Arenal, Zapopan 45019, Jalisco, Mexico; asuarez@ciatej.mx

*   Correspondence: ncruz@uaeh.edu.mx

**Abstract:** The growing interest of consumers to find products with greater health benefits has led to multiple research works focused on product developments with antioxidant-rich foods by creating safe, acceptable, and high-value nutrition, even in those foods susceptible to deterioration, in order to replace synthetic preservatives with natural ones. Bee products are an alternative addition for food products (dairy and meat products, coating fruits, and alcoholic and non-alcoholic drinks), which can improve the final quality of the food for the benefit of the consumer. The aim of this review was to provide detailed information on the main findings of innovative food products based on the addition of bee products by highlighting their physicochemical and functional properties and their behavior throughout storage.

**Keywords:** bee products; antioxidant-rich; new product developments; functional properties

## 1. Introduction

The history of bee products dates back to ancient times; for example, the Greeks believed that pollen and honey were responsible for giving youth and life to kings, and as a result, it was highly valued. Further, it also performed an important role in religious rites [1,2].

Honey is the main product generated by bees and recognized for its sweet flavor derived from the nectar of flowers that bees collect, transform, and combine with specific substances of their own, and is then stored in the honeycomb to mature. It is essentially composed of sugars, predominantly fructose and glucose, in addition to other substances such as organic acids and enzymes [3]. Another apiary product is pollen, which is a natural product of great biological value obtained from hives.

The composition of bee products have been widely discussed depending on the geographical origin, as well as the importance of quality through the years [4]. In general, they are constituted by proteins, lipids, and vitamins, (mainly vitamin D), flavonoids, minerals (iron and zinc), dietary fiber, and carotenoids; however, this depends on drying techniques and storage conditions of the pollen [5]. In dairy products (yogurt, milk, and cheese), amins, pigments (chlorophylls, carotenoids), and aminoacids can act together as antioxidants [2,6].

Propolis is a solid collected by bees from the buds and exudates of plants, mixed with bee enzymes, pollen, and wax. It has been recognized in alternative medicine for its antimicrobial, anti-inflammatory, antitumor, immunomodulatory, and antioxidant activities [7,8]. Finally, there is the royal jelly, which is a thick, milky discharge produced by the hypopharyngeal and mandibular glands of bees. It serves as food for the larvae during the first three days and for the queen bee [9].

In recent years, multiple scientific research works maintain that dietary patterns have specific effects on the health–disease binomial, related to lifestyles, malnutrition and sedentary behavior, resulting in an increase in mortality [10]. As a consequence, the industrialized world faces new challenges, where researchers place emphasis on improving scientific knowledge about alternatives to achieve *"optimal nutrition"*, which can be defined as a nutrition model with nutrients or bioactive compounds to prevent disease and protect health, derived from the growing consumer demand for high-quality life standards [11]; therefore, enrichment is an option to provide micronutrients and health-promoting compounds in processed foods [12].

This approach has generated interest in taking advantage of the properties of apiary products in the food industry. In this paper, our objective was to present an overview of new food product developments with bee-related products added (honey, pollen, royal jelly, and propolis) and their impact on the food's physicochemical characteristics and functional value. The review has been prepared considering six food product categories: dairy, fruits, drinks, meat, bakery and related products enriched with apiary products.

## 2. Dairy Products

The effect of apiary products content on the survival of yogurt starter bacteria, storage time, temperature, nutritional and functional composition, and sensory properties of dairy products (yogurt, milk, and cheese) is shown in Table 1.

**Table 1.** Effect of the concentration of bee products in dairy products, on bacterial cultures and physicochemical properties.

| Apiary Product | Origin | Botanical Origin | Content Added | Type of Product | Storage Temperature (°C) | Storage Days | Bacterial Culture | Effect | Reference |
|---|---|---|---|---|---|---|---|---|---|
| Honey | Egypt | Fennel honey | 5, 10, 15% | Yogurt | 6 ± 2 | 0, 3, 7, 14 | ABT-5 cultures (*Lactobacillus acidophilus*, *Bifidobacterium bifidum* and *Streptococcus thermophilus*) | Increase in Bifidobacteria counts during and at the end of storage time (until $54 \times 10^6$/g CFU), titratable acidity and total soluble solids were proportional to the honey content added and the storage time. | [13] |
| Honey | Egypt | —— | 0, 2, 4, 6% | Yogurt | 4 | 0, 7, 15 | *S. thermophilus*, *L. delbrueckii* subsp. *bulgaricus* (1:1) and ABT-5 | Bifidobacteria counts, acceptable to exhibit probiotic effect ($1.7–3.5 \times 10^6$ CFU/g, at day 15), good sensory attributes, in particular, sample with 5% honey. | [14] |
| Honey | Egypt | Sedr honey | 5, 10, 15, 20% | Yogurt | 6 ± 2 | 0, 3, 7, 14 | *L. delbrueckii* ssp. *bulgaricus*, *S. thermophilus* and ABT-5 | Increases in *B. bifidum* and *S. thermophilus* and high protein proportion compared to the control, | [15] |
| Honey | Egypt | —— | 5% | Yogurt | 4 | 0, 7, 14 | *S. thermophilus* and *L. delbrueckii* subsp. *bulgaricus* (1:1) and ABT-5 | High *L. bulgaricus* and *S. thermophilus* counts, significant increase in ashes, unsaturated fatty acids content, improvement of consistency and flavor. | [16] |

**Table 1.** *Cont.*

| Apiary Product | Origin | Botanical Origin | Content Added | Type of Product | Storage Temperature (°C) | Storage Days | Bacterial Culture | Effect | Reference |
|---|---|---|---|---|---|---|---|---|---|
| Honey | Turkey | Pine | 3, 5, 7% | Yogurt | 4 | 0, 7, 14, 28 | *S. thermophilus* and *L. delbrueckii* subsp. *bulgaricus* (1:1) | The addition of 7% honey presents the highest *L. bulgaricus* counts (8.27 log CFU/g). Better water retention capacity (led to lower syneresis), high viscosity than the control, low water activity, enhancement of antioxidant properties. | [17] |
| Honey | Romania | —— | 7% | Yogurt | —— | —— | *S. thermophilus*, and *L. bulgaricus* | Increase in *S. thermophilus* counts during storage, and a decrease in *L. bulgaricus* at the end of storage. | [18] |
| Honey | Brazil | —— | 5, 10, 15% | Yogurt | 4 ± 2 | 1, 7, 14, 21, 28 | *S. salivarius* subsp. *thermophilus* and *L. delbrueckii* subsp. *bulgaricus* and the probiotic culture of *L. acidophilus* La-05 | Honey as a growth-promoting probiotic ingredient (most evident between day 14 and before day 21 of storage). Odor characteristics, syneresis, viscosity, water retention capacity, sensory acceptance, and purchase intention were positively affected. | [19] |
| Honey | Turkey | Pine | 2, 4, 6% | Yogurt | 4 | 1, 7, 14, 21 | *S. thermophilus*, *L. delbrueckii* subsp. *bulgaricus*, *L. rhamnosus*, *L. acidophilus*, *L. plantarum*, *B. animals* subsp | *L. acidophilus* and *L. delbrueckii* counts were minor to the control. The presence of honey increases acidity, promote less syneresis and color modification with increasing concentration (*L*\* values were affected). | [20] |
| Honey and royal jelly | Egypt | Egyptian clover (honey) | Honey: 2, 4, 6, 9% Royal jelly: 0.2, 0.4, 0.6, 0.8, 1.0, 1.2, 1.5% | Yogurt | 5 ± 1 | 0, 3, 6, 9 | *S. thermophilus* and *L. delbrueckii* subsp. *bulgaricus* | Improves viability of *S. thermophilus* with an addition of 4% honey, suggesting a prebiotic effect. An increase in viscosity and mineral content was observed. | [21] |
| Honey | Algeria | Multifloral | Honey: 0, 2.5, 5% Pomegranate peel: 0, 2.5, 5, 10% | Yogurt powder | 4–6 | —— | *S. thermophilus* and *L. delbrueckii* subsp. *bulgaricus* | No significant differences in moisture content or particle sizes distribution and density values were affected. Fortified samples showed higher phenolic content and antioxidant activities than control samples. | [22] |
| Honey | Slovakia | Rape honey | 1, 3, 5% | Yogurt | 6 ± 1 | 1, 7, 14 | Laktoflora® | The dry matter content increases gradually with the addition of honey, with a lower cohesion and firmness, but a high antioxidant activity. | [23] |

**Table 1.** *Cont.*

| Apiary Product | Origin | Content Added | Type of Product | Storage Temperature (°C) | Storage Days | Bacterial Culture | Effect | Reference |
|---|---|---|---|---|---|---|---|---|
| Bee pollen | Turkey | 2.5, 5.0, 7.5, 10, 20 mg/mL | Fermented milk beverages | 4 | 1, 7, 14, 21 | *L. acidophilus*, *Bifidobacterium animalis* subsp. *lactis* and *S. thermophilus* (ABT-1) | Possible antimicrobial effect against *B. Lactis*, negative effect on sensory properties. Increase in soluble solids content, and viscosity, more proteolytic activity (first day). | [24] |
| Bee pollen | Egypt | 0, 0.5, 1.0, 1.5, 2.0% | White cheese (camel and cow milk) | 10 | 0, 15, 30, 45 | — | Antibacterial activity (with 2%) for *S. typhimurium* and *E. coli*, 31 times high in total phenolic content (till 46.12 mg/g), detrimental effect on sensory attributes (texture, taste, acceptance, odor). | [25] |
| Bee pollen | Greece | 0.5, 1.0, 2.5, 3.0% | Yogurt (cow, goat and sheep) | 4 ± 1 | — | *S. thermophilus* and *L. bulgaricus* | Radical inhibition around 100%, the potential effect of the surface material causing an improvement in appearance and cohesion. | [26] |
| Bee pollen | India | 5, 10, 15%, with variations in temperature and pressure | Milk powder | 4 | — | — | Negative effect on solubility, apparent density within the limit for powdered foods, improvement of a lower molecular weight sugar concentration, resulting in a more hygroscopic powder. | [27] |
| Propolis | Egypt | 1, 2, 3% (aqueous extract) | Yogurt | 5 ± 1 | 0, 7, 14 | *L. delbrueckii* subsp. *bulgaricus* and *S. thermophilus* (1:1) | Variations in coagulation time were proportional to the aqueous extract of propolis added (synergistic effect); samples with 1 and 2% represented the highest sensory scores (fresh and during storage). | [28] |
| Propolis | Brazil | 0.05% | Yogurt | 4 ± 2 | 0, 7, 14, 28 | *L. acidophilus*, *Bifidobacterium* and *S. thermophilus* | Adequate concentrations of oleic and linoleic acid, without negative interactions in the survival of lactic acid bacteria, and high levels of antioxidant activity. | [29] |
| Propolis | Brazil | 0.05% | Yogurt | 4 ± 2 | 0, 7, 14, 21, 28 | *L. acidophilus*, *Bifidobacterium* and *S. thermophilus* | Inhibitory effect for *Salmonella* spp., and *E. coli.*, cohesion increases, good sensory acceptance. | [30] |
| Propolis | Brazil | 0.5, 1.0, 1.5, 2.0% | Commercial milk, yogurt and Kefir | 4 | — | *S. thermophilus*, *L. delbrueckii* subsp. *bulgaricus*, *L. acidophilus*, and *Bifidobacterium animalis* subsp. *lactis* | Commercial milk, yogurt, and Kefir supplemented with 0.5% of propolis resulted in best organoleptic characteristics for each product. | [31] |

**Table 1.** *Cont.*

| Apiary Product | Origin | Content Added | Type of Product | Storage Temperature (°C) | Storage Days | Bacterial Culture | Effect | Reference |
|---|---|---|---|---|---|---|---|---|
| Bee pollen and royal jelly | Egypt | Royal jelly: 0.6% Bee pollen: 0.8% | Yogurt | ~5 | 0, 7, 14, 21 | *L. delbrueckii* subsp. *bulgaricus* and *S. thermophilus* (1:1) | Decrease of *S. thermophilus* and *L. delbrueckii* subsp. *bulgaricus* counts. Total content of solids, ash, fat, protein, and acidity increases significantly. Acceptability improving during storage (up to day 7). | [32] |
| Royal jelly and bee pollen | Egypt | Royal jelly: 0.6% Bee pollen: 0.8% | Yogurt | 4 ± 1 | 0, 7, 14, 21 | *L. delbrueckii* spp. *bulgaricus*, *S. thermophilus*, *Bifidobacterium angulatum*, *L. rhamnosus*, and *L. gasseri* | Refrigeration conditions increase hardness and chewiness in bee product added samples, resulting in a better texture and less syneresis. | [33] |
| Royal jelly and bee pollen | Egypt | Royal jelly: 0.6% Bee pollen: 0.8% | Yogurt | 4 ± 1 | — | *B. angulatum*, *L. gasseri* and *L. rhamnosus* | Increase of amino acids and fatty acids content, high presence of acetaldehyde in treatment with 0.6% of royal jelly. | [34] |
| Honey and bee pollen | Bulgaria | Honey 5, 10, 15% Bee Pollen: 0.4, 0.6, 0.8% | Yogurt | — | — | *S. thermophilus* and *L. delbrueckii* ssp. *Bulgaricus* | No changes were observed in the organoleptic properties, with the incorporation of honey in 5% and pollen in 0.4%. | [35] |

There is significant interest in adding honey to dairy products because, in addition to being considered a sweetener, it produces changes in several characteristics (bacterial survival, nutritional composition, antioxidant content, antioxidant activity, and sensory properties). A similar effect has been observed in pollen, royal jelly, and propolis addition; thus, these changes are described below:

*2.1. Bacterial Survival*

The maintenance of bifidobacteria survival in milk is a challenge for the food industry due to the high nutrient demands of bacteria as well as proteolytic and glycolytic reactions [36]. It is reported that the oligosaccharides present in honey exhibit a potential prebiotic effect, stimulating the probiotic bacterial growth, thus producing an increase in bifidobacteria and lactobacilli counts [15,37]. A formulation with 5% of honey maintains the viability and average counts for *L. acidophilus*, even during storage (21 days) [17]. Likewise, in a yogurt powder, the incorporation of honey (2.5%) did not affect the lactic acid bacteria counts, finding similar CFU (Colony Forming Units) values to the control ($3.1 \times 10^8$ and $2.8 \times 10^8$ CFU/g, respectively) [22]. Further, bee pollen has a positive effect on bacterial survival for *L. acidophilus* in fermented milk beverages; in contrast, when pollen was incorporated at high concentrations (20 mg/mL) [24]. This difference in results may be due to the *L. acidophilus* surviving in acid conditions [38].

*2.2. Nutritional Composition*

In general, the chemical composition of dairy products is determined by the bacteria metabolic activity that interacts with the environment during their growth, generating certain components as result of their metabolism [39]. The honey addition to yogurt samples causes changes during storage, resulting in an increase of 7% in fat and 25% in protein content [13].

There is a possible explanation for the protein increase, as it has been reported that honey contains a small concentration of protein [40,41]; however, this depends on the amount of honey added as well as its quality. With respect to fat, there is some controversy

over whether honey leads to an increase in fat content; because other studies maintain that adding honey to yogurt leads to a decrease in fat content [30], and that honey has a low or even no fat content [42].

Regarding mineral profile, the honey incorporation between 2–9% showed high mineral content in comparison to control because apiary products are considered a rich source of minerals [21].

When added to cheese samples, pollen significantly increased the amount of protein in cheese samples [25]. Further, the ash concentration in yogurt increased considerably, which was related to its high dry matter concentration [32]. In relation to aminoacids and fatty acids content, yogurt with 5% honey added, as well as coconut milk, causes a decrease in saturated fatty acid content, but in medium and long-chain fatty acids, the content was similar to the control yogurt [16].

When pollen (0.6%) is present in yogurt, it provides a significant amount of valuable amino acid content (isoleucine, histidine, lysine, valine, methionine, threonine, arginine, proline, and cysteine). Furthermore, this addition resulted in a product with high amounts of fatty acids, except for butyric acid [34]. This effect could be due to the diversity of aminoacids (threonine, phenylalanine, and leucine) [43] and fatty acids (palmitic acid, stearic acid, oleic acid, linoleic acid, linoleic acid, and eicosenoic acid) contained in pollen [44], both fatty acids and aminoacids are important compounds for human health. Finally, the addition of propolis into yogurt generated similar fatty acid content compared to the control [29].

### 2.3. Physicochemical Properties

Some authors have dedicated efforts to determine how apiary products modify the pH and titratable acidity in yogurt; they affirm that 2% to 7% of added honey causes a decrease in acidity [14,17]. It is possible that the fructooligosaccharides contained in honey are responsible for this effect, where its addition could modify the pH of the products [37]. In other research work regarding acidity, the incorporation of pollen into yogurt leads to low levels of titratable acidity, even at minimal proportions of 0.4, 0.6, and 0.8% [35]. This is probably caused by the antimicrobial effect of pollen on *S. thermophilus* and *L. bulgaricus*, leading to a harmful effect on lactic acid bacteria [45]. On the other hand, the propolis extract in raw milk resulted in significantly high acidity values, which could be attributed to the improvement of lactic acid bacteria action, leading to a greater decomposition of sugars in the milk and an increase in acidity [28]. Numerous studies concluded that honey generates changes in total soluble solid content when it is added to dairy products [14,16,17,24]. It can be due to the fact that honey is composed mostly of sugars (they comprise approximately 95% of the dry weight), resulting in a high concentration [46].

Color is an important quality attribute of yogurt related to its acceptability, and it is considered one of the first attributes perceived by consumers [47]. The brightness (*L\**) of yogurt is related to the size of the fat and protein particles, which can affect their ability to disperse and reflect light [48]. In this way, the incorporation of honey into dairy products produces an increase in the *a\** values; however, these values depend on the amount of honey added. In yogurt, a 7% honey content generates a reduction in the brightness (*L\**) values [17].

With respect to rheological properties, the addition of honey caused an increase in the viscosity values (up to 126%) [17]. In comparison, the propolis addition promoted a non-Newtonian and pseudoplastic behavior [30]. In fermented beverages, the addition of pollen also allowed a significantly high viscosity [24].

Regarding the microstructure of dairy products, a yogurt formulation with added pollen resulted in a product with an uniform distribution and size of the casein micelles, giving rise to a product with more consistency and syneresis control during its storage [33].

### 2.4. Antioxidants Content and Antioxidant Properties

The addition of honey into yogurt significantly increases the total phenolic content, maintaining the differences with respect to the control yogurt during 4 weeks of storage [13]. Furthermore, 5% of honey in yogurt caused high values of antioxidant activity (5 more times) compared to the control [21]. Further, propolis extract caused a considerable increase in phenolic, flavonoids content, and antioxidant activity (aproximately 52%, 51%, and 4.54% more, respectively) than control yogurt [49]. Likewise, in buffalo milk powder with 20% of honey added, the phenolic compounds and flavonoids content were 5 and 28 times higher, respectively, as well as the antioxidant activity increased up to 12.61%, in comparison with the control sample [50]. Finally, a process of milk with pollen added was optimized, where the optimal pollen amount to preserve the functional properties was 13.72%, with a processing temperature of 26.84 °C, and 23.37 in Hg pressure [27]. The increase in polyphenol content and antioxidant activity is due to the fact that apiary products are rich sources of antioxidants [51], generating a high content when they are added to yogurt and milk.

### 2.5. Sensory Evaluation

Honey is recognized for its sweet taste, so when it is incorporated into other foods, it is necessary to evaluate whether it really improves the sensory properties. A level of 5% honey in yogurt promoted better acceptance in terms of taste, texture, and consequently in the sensory attributes of yogurt, even after being stored for 14 days [13]. Further, with honey levels between 5%, and 7%, it does not affect the level of acceptance, consistency, nor sweetness [17]. When pollen was added to yogurt (0.5%), it resulted in a product with a high acceptance level; however, when the total levels were higher than 1%, the flavor and taste were affected [26]. On the other hand, in fermented beverages, the addition of pollen between 2.5–20 mg/mL negatively influenced the taste, but on the contrary, a positive impact was observed in the texture scores. This effect could be due to the increase in viscosity by increasing the dry matter content, causing an improvement of the structural and sensory properties [24].

The addition of royal jelly into yogurt (0.6%) resulted in a product with good sensory quality, without a negative effect on the survival of lactic acid bacteria [21,32]. The propolis (1% and 2% in ethanolic extract) mixed with raw milk produced better characteristics in terms of aroma, body, texture, flavor, and general acceptability [28]. Further, a yogurt with 0.05% of royal jelly generated pleasant sensory characteristics. Nevertheless, formulations contained corn syrup and flavorings, and this could have favored the result [30].

Bee products have also led to several beneficial effects on products susceptible to alteration during storage. These results are directly related to each bee product; the honey and bee pollen provide sweetness and fruity taste [52,53], royal jelly provides a spicy taste [54], and the propolis adds a toasted, sweet, nutty taste [55]; however, these results depend on the amount added.

Their implementation in fruits can protect the fruits against deterioration; this implementation is described in the following section.

## 3. Fruits

Propolis also has a diversity of applications in food technology, highlighting its use in the improvement of some attributes of fruits through the generation of biodegradable coatings, which can be another option for the production of films [56,57]. Artepillin C is a compound of propolis that has beneficial effects, such as the preservation of fruits and a powerful antifungal activity [58].

For example, banana is a climacteric fruit that ripens quickly, losing quality and sensory attributes over time [59]. In response to this, the food industry has implemented new alternatives that allow regulating their deterioration. Propolis extract has been incorporated (2.5%) in bananas to prevent weight losses during a storage period of 12 days, although negative changes in the acceptance of the product were found [60]; however, when 5% of propolis extract was applied in orange peel, as a protective coating, the storage period was

extended up to ten weeks [61]. The incorporation of 40% of honey in amla papaya jam generated a product with a high content of total soluble solids, carbohydrates, and ashes, and also increased their acceptability [62].

Another important application of bee products is in beverages; this is described in the following section.

## 4. Enriched Drinks

On a global scale, fruit juices are a versatile option that consumers find as an important part of the modern diet for their nutritional properties and sweet taste. Hence, their preservation is an important subject for food researchers [63]. The propolis addition (0.02 mg/mL) to juices shows an ability to maintain pH and titratable acidity in a period of 5 weeks, maintaining the carotenes and the color (measured as luminosity parameter) stability. It can also be an alternative to preserve the vitamin C content against degradation during storage (the propolis generates 13.12% less oxidation of vitamin) [64].

Likewise, in citrus juice samples, the addition of pollen (0.25 and 0.60%) increases the total phenols content (26.7% more), as well as the antioxidant activity. In contrast, in a sensory evaluation, its presence modifies the color and produces a sensation of powder on the tongue, identified by some panelists as unpleasant. Despite these changes, more than 70% of the panelists indicated that they would buy the product [65].

In malt beverages, pollen addition increases the content of phenols and flavonoids by 45.7% and 211.6%, respectively [66]. Finally, pollen in drinks improves not only the antioxidant properties but also its sensory characteristics. In wines with pollen concentrations of 0.1 and 0.25 g/L, it promotes better aromatic and taste characteristics; however, a high proportion is related to a lower aroma score due to a typical or similar odor of nuts or vegetables, accompanied by notes of bitter taste [53].

On the other hand, the quality and durability of meat products are a challenge in the food industry; therefore, bee products have shown interesting effects on their properties; this is described in the following section.

## 5. Meat Products

Meat products and all foods of animal origin are considered very susceptible to being affected by microorganisms. The addition of propolis ensures microbial stability and food quality during storage [67].

In general, the fat content and composition profile in hams and sausages play a fundamental role in their technofunctional characteristics; however, during processing, lipids are vulnerable to oxidation affecting their structure and generating the presence of various harmful volatile compounds, such as aldehydes, which are responsible for meat products becoming rancid. Lipid oxidative stability is a challenge for the food industry [68]. The main findings regarding the effects of bee pollen and propolis addition regarding lipid oxidation, physicochemical composition, microbiological characteristics, and sensory quality are described in Table 2.

**Table 2.** Effect of the addition of bee products as natural antioxidants in storage conditions, on lipid oxidation, composition, microbiological, and sensory quality in meat products.

| Apiary Product | Origin | Content Added | Type of Product | Storage Temperature (°C) | Storage Days | Effect | Reference |
|---|---|---|---|---|---|---|---|
| Propolis | Slovakia | 0.06% in extract | Cured cooked ham | 4 | 21 (sliced) and 20, 50, 100 (unsliced). | Lower TBA values in unsliced hams after 100 days of storage. Hams sliced with propolis have a lower aroma intensity. | [69] |
| Propolis | Italy | Powdered to 5% | Fish burgers | — | — | Formulation with 5% spray-dried propolis and 10% potato flakes and 9% olive oil, shows an increase in antioxidant activity and better sensory quality. | [70] |
| Propolis | Colombia | 0.8% | Sausages with 60% porcine meat, 20% bovine meat and 20% porcine fat | 50 | 0, 8, 16, 24 | TBA value similar to sausage with sodium nitrite, lower concentration of volatile nitrogenous bases in all storage, the propolis addition does not modify the consumer acceptance. | [71] |
| Propolis | Colombia | 0.8, 1.2% | Fish fillets | 3 | 0, 4, 8, 12, 16, 20, 24 | Growth inhibition of bacterial pathogens (*Clostridum* sp., *Salmonella*, *Escherichia coli*, *Staphylococcus aureus*), acceptable microbiological load in storage with 0.8 mg/mL and 1.2 mg/mL (6.1 log CFU/g and 5.4 log CFU/g, respectively), color remained unchanged at 0.8% propolis added. | [72] |
| Propolis | Brazil | 0.01, 0.05% and compared to TBH | Italian-type salami (pork meat) | 18 | 1, 15, 35 | No relevant changes in proximal composition, protective effect similar to BHT or TBAR with the extract of 0.05% propolis. | [73] |
| Propolis | Brazil | 0.1 g/Kg | Burger meat (lean beef) | $-15 \pm 0.6$ | 0, 7, 14, 21, 28 | Effective protective agent to control TBA values similar to sodium erythorbate, aroma and flavor results below ideal. | [74] |
| Propolis | China | 1% | Fish sausages | 2 | 63 (9 weeks) | TBA values below the limit of 5 mg/kg in fish meat and lower than the control sample during storage, an improvement in shelf life is concluded, reduction in sensory attributes. *Escherichia coli* and *Salmonella* were not detected in any sample. | [75] |
| Bee pollen | Turkey | 0, 1.5, 3.0, 4.5, 6.0% | Frozen meatballs | $-20 \pm 1$ | 0, 30, 60, 90 | Lower TBA values, total viable counts decreases. Chroma (C) and hue angle (h) values increases with 4.5 and 6.0%, respectively as the concentration of pollen is added. | [76] |
| Bee pollen | Brazil | 0.2 g/kg | Pork sausages | 4 | 0, 5, 10, 15, 20, 25, 30 | Control of lipid oxidation exhibiting lower TBA values, compared before (0.67 vs. 2.44 mg/kg) and after storage (4.08 vs. 4.71 mg/kg). | [77] |
| Bee pollen | Brazil | 0.1 g/Kg | Beef burger | $-12$ | 0, 7, 14, 21, 28, 35 and 42 | Lower TBA values in the samples with pollen, suggesting an increase shelf life up to 42 days storage. | [78] |

The addition of propolis extract (0.06%) in cured ham is an alternative method to control lipid oxidation, producing a decrease in odor [69]. A similar protective effect was found in salami samples (0.05% propolis extract) in comparison to the positive control by using BHT (butylated hydroxytoluene) after 35 days of aging [73]. Further, burger meat with microencapsulated propolis (0.1 g/kg) added showed a better lipid oxidative stability than the use of a commercial additive (sodium erythorbate). The burger meat with added propolis showed a lower TBA (thiobarbituric acid) value than control during storage. The maximum malonaldehyde value was reached in 14 days, whereas in the control sample, it was reached at 7 days. This could be due to the gradual release of

propolis bioactive compounds; however, for this product, the sensory characteristics and the general consumer acceptance were negatively affected [74].

Similarly, in fish burger meat with 5% propolis, changes affecting the odor were observed [70]. This effect was also observed with 1% propolis extract addition in fish sausages; however, in this research, it was concluded that the addition of propolis extract increases the shelf life of fish sausages by up to 3 weeks [75].

Definitively, the lower acceptance scores for meat with propolis addition could be related to their own taste and odor characteristics affecting the final product acceptance [74]. Although a perception of astringency caused by phenolic compounds is possible, depending not only on the amount added, but also on the composition of the enriched food [79].

On the other hand, 0.2% bee pollen in pork sausages shows a greater antioxidant effect and a better lipid peroxidation control during 30 days of storage by inhibiting the oxidation in 13.37% when compared to the control [77]. Further, in meatballs with added pollen, the lowest values of malonaldehyde were found at 90 days of storage. These results indicate that pollen is effective in delaying lipid oxidation in meatballs [76]; this mechanism could be associated with the presence of cinnamic acid derivatives and flavonoid content [80].

Some additives such as nitrate (antioxidant and preservative) are used in meat products; however, it is known that it has detrimental effects, e.g., reacting with the biogenic amines by the decarboxylation of some aminoacids [81], and it produces nitrogenous compounds known as cancer precursors [82]. Thus, propolis is a promising natural substitute with antimicrobial agents. When propolis was added in chorizo samples at 0.8 mg/mL, the growth of some bacteria, such as *Staphylococcus aureus*, *Escherichia coli*, *Salmonella* subsp., and *Clostridium* subsp., was inhibited. Furthermore, the addition does not cause any modification in the sensory attributes [71]. A 0.8% of propolis extract was used on fish fillet samples, resulting in a similar effect against the same bacteria previously described; an improved color score was also observed [72], where phenolic acids, anthraquinones, and flavonoid content [83], are thought to be responsible.

Currently, research has focused on the incorporation of bee products for the nutritional improvement or substitution of ingredients in formulations of baked products. This addition is considered to be a reasonable alternative to traditional additions, has better characteristics, and has a high acceptance rate by consumers; the addition of bee products to baked goods is described in the following section.

## 6. Bakery and Related Products

Bakery products are among the most popular ready-to-eat food products in the world [84]. Currently, a large proportion of people suffer intestinal malabsorption and gluten is often excluded from their diet [85]. Researchers have designed some gluten-free bakery products as an option; however, these products have a lower nutritional quality. To counter the lower nutritional quality, they must be fortified with other ingredients, e.g., amaranth, quinoa, and other seeds with high protein content [86–88]. Adding pollen in the range of 5% to gluten-free bread increased the content of protein by 0.54%, the potassium by 20%, and the calcium by 37%. The addition also improved the bioaccessibility of antioxidant compounds with an increase of 36% and the concentration of phenolic compounds by up to 11.2% [89].

In cookies with formulations containing 5.0%, 7.5%, and 10% of pollen, a significant increase in the protein content was obtained (0.6% more), a high percentage of ashes (up to 0.59%), and lower $L^*$ values. The lower $L^*$ values can be explained by the darkening caused by the pollen; however, it results in an increase in the total phenolics content by up to 2.9 and 2.3 more times that of the antioxidant activity. Finally, a positive effect on the maintenance of sensory parameters, by adding a concentration of 5%, since a greater amount produces changes in the taste, making it bitter and with a less pleasant consistency [90].

On the other hand, the mineral profile of a 15% honey enriched bread showed a calcium and iron content up to 12.53% and 35.34% more, respectively [91].

In cookie samples, with the addition of pollen in 16% and 32% (equivalent to 1 or 2 g), the protein content ranged from 7.18–9%; regarding ash, the sample with 16% contained the maximum antiradical activity with 69.91%, which leads to a pleasant taste, since high amounts (proportion higher than 16%) caused a decrease in acceptance, odor, and flavor with a predominance of cabbage notes [92].

Finally, the propolis addition (1%) in a croissant with honey (25%) caused an improvement in the color scores, without generating significant changes in the odor; however, an addition greater than 1% impairs the acceptance, decreases the moisture, but increased the ash and protein content (up to 1.17% and 19.79%, respectively) [93].

## 7. Enrichment of Apiary Products

Bee products have been used since ancient times in popular medicine due to their wide variety of beneficial health properties [94–97]. The number of bioactive compounds could be the basis for the attention, particularly for formulation development and food innovation [98]. Studies on the physico-chemical properties of apiary products with mixture of other bee products are scarce; due to most studies focus on the addition of honey, polen, propolis or royal jelly to other food matrices. Honey enriched with propolis at 0.1–1% intervals no generates changes in humidity, fructose, glucose, or sucrose content. It was observed that propolis added in the largest amount (1%) leads to an increase in the total phenolic, flavonoids, total phenolic acids, anthocyanins, and carotenoid content of 4.34, 5.37, 3.98, 2.61, and 1.37 more times, respectively. Further, in the same research, the same addition exhibited chrysin and *p*-cumaric acid as major compounds, reaching levels of 775 and 179 more times, respectively, compared to the control. Finally, regarding sensory properties, the propolis did not present changes in acceptance, except in a proportion higher than 0.3%, which results in low acceptability (unpleasant sensation and bitter taste), and the presence of an aftertaste as well as a change in the color and odor [99].

Other evidence establishes that honey enriched with propolis extract (0.3% and 0.5%, with 90% ethanol and 48 h of maceration) is promising, as it can maintain a good acceptance level in consumers. Furthermore, it showed a total phenolic compound and high flavonoids content (270.08 mg/100 g and 15.68 mg/100 g, respectively) compared to honey; however, the mixture with more than 0.5% of the extract was perceived as unpleasant, bitter, and intense [100].

A mixture of honey with dry cherries (40%) had better acceptability scores, even when stored for 3 months. Moreover, the total phenolic and flavonoid content increase up to 2.17 and 2.81 more times, respectively; this could be due to the fact that cherries are a rich source of antioxidants [101]. Similarly, a combination of honey with bee bread (10–60%) resulted in a product with a significant total of phenolic, flavonoids content, and an antioxidant activity of 3.99, 0.3, and 4.08 more times, respectively, than the control sample [102]. Furthermore, a mixture of honey with propolis extract (1%) and bee bread (15%) resulted in a product with a high phenolic compounds content (150 mg/100 g) and a high antioxidant activity (6.5 more times) as evaluated by DPPH$^{\bullet +}$ [103].

## 8. Conclusions

Given the importance of bee products described in this review (honey, pollen, propolis, and royal jelly) and the great variability of food matrices to which they can be incorporated, it is important to highlight the potential benefits as food ingredients. In dairy products, honey improves sensory attributes such as taste and texture, and results in an increase in phenolic compounds and antioxidant activity; however, viscosity increases and luminosity values decrease, which is seen as unfavorable. Propolis is an effective ingredient for the production of edible fruit films, and in sausages and meat products, it generates protective effects similar to synthetic preservatives due to its antimicrobial activity. Pollen addition in cookies leads to a significant increase in some micronutrients, as well as fiber, protein, phenolic compounds, and antioxidant activity. With respect to royal jelly, its incorporation

into dairy products improves the physical and chemical characteristics during storage; it also allows for a uniform distribution of the casein micelles.

We recommend that future research for the food industry should be to find the optimal concentration of bee product additives (as ingredients) for different food matrices, without forgetting that, for every food product development, e.g., antioxidant-rich foods, it is necessary to have a safe final product, which provides health benefits to consumers, as well as good sensory properties.

**Author Contributions:** Investigation, G.I.C.-B.; Writing—Original Draft Preparation, G.I.C.-B., N.d.S.C.-C., and E.R.-M.; Writing—Review and Editing, N.d.S.C.-C., E.R.-M., A.C.-O., Q.Y.Z.-R., Á.S.-J., and L.D.-O. All authors have read and agreed to the published version of the manuscript.

**Funding:** The first author was supported by a fellowship from the Consejo Nacional de Ciencia y Tecnología (CONACYT) (No. 764080).

**Institutional Review Board Statement:** Not applicable.

**Informed Consent Statement:** Not applicable.

**Data Availability Statement:** Not applicable.

**Conflicts of Interest:** The authors declare no conflict of interest.

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
