# Peer review of "Addition of Bee Products in Diverse Food Sources: Functional and Physicochemical Properties"

_applsci, doi:10.3390/app11178156_

Round 1
Reviewer 1 Report
The paper ”Addition of Bee Products in Diverse Food Sources: Functional 2 and Physicochemical Properties” summarizes the progress in the addition of apiary products, specifically honey, pollen, royal jelly, and propolis, to various food matrices and the physicochemical and functional value modifications that come with. It is a good paper, well written, containing valuable information for both industrials and food science researchers. There are some aspects that need to be added.
The abstract is too general. Please refer to the addition of bee products into food.
Section 2 might be eliminated and the other sections to be renumbered. Obviously, you searched all the international databases and for sure you did not use only those searching keywords.
I suggest avoiding the split of lines 82 and 97 by table 1 as it is difficult to follow.
Table 1: Please look also at the recent papers focusing on the addition of bee products into dairy products to have a better map distribution of citing references. Here are 2 examples:
Brazil - https://doi.org/10.1016/j.lwt.2017.02.013
Turkey - https://doi.org/10.1016/j.lwt.2021.111444
Brazil, Korea - https://doi.org/10.22424/jdsb.2020.38.2.59
Some spelling errors were encountered. Please check carefully the entire manuscript.
Lines 201-206 are misunderstanding. Please rephrase.
References: the number of citing references is low. Recent papers were not cited in this review paper:
Fortification of beef burger with the addition of bee pollen from Apis mellifera L. - https://doi.org/10.9755/ejfa.2019.v31.i11.2025
The application of pollen as a functional food and feed ingredient—the present and perspectives - https://doi.org/10.3390/biom10010084
Composition and functionality of bee pollen: A review - https://doi.org/10.1016/j.tifs.2020.02.001
I agree with the publishing of this paper after minor revision.
Author Response
Mexico, 24 august, 2021
We would like to thank reviewers for the comments made to our manuscript entitled “Addition of Bee Products in Diverse Food Sources: Functional and Physicochemical Properties”. We include our responses to the comments raised by the reviewers point by point, as well as, revision of all manuscript, spelling errors and the English language by a native speaker. Corrections and added information have been shaded with yellow.
REVIEWER 1
The paper ”Addition of Bee Products in Diverse Food Sources: Functional 2 and Physicochemical Properties” summarizes the progress in the addition of apiary products, specifically honey, pollen, royal jelly, and propolis, to various food matrices and the physicochemical and functional value modifications that come with. It is a good paper, well written, containing valuable information for both industrials and food science researchers. There are some aspects that need to be added.
The abstract is too general. Please refer to the addition of bee products into food.
Answer: The information has been added in line 17-19
Section 2 might be eliminated and the other sections to be renumbered. Obviously, you searched all the international databases and for sure you did not use only those searching keywords.
Answer: The methodology section has been removed as suggested and renumbered
I suggest avoiding the split of lines 82 and 97 by table 1 as it is difficult to follow.
Answer: The default template provides a line in the header and footer, however, the intermediate lines of the tables have been removed
Table 1: Please look also at the recent papers focusing on the addition of bee products into dairy products to have a better map distribution of citing references. Here are 2 examples:
Brazil - https://doi.org/10.1016/j.lwt.2017.02.013
Turkey - https://doi.org/10.1016/j.lwt.2021.111444
Brazil, Korea - https://doi.org/10.22424/jdsb.2020.38.2.59
Answer: The suggested papers have been added to Table 1, however Turkey paper it was not possible to obtain it so another reference was added in the table 1. Which we present below:
Coskun F, Dirican LK. Effects of pine honey on the physicochemical , microbiological and sensory properties of probiotic yoghurt. 2019;2061:616–25 (Turkey)
Some spelling errors were encountered. Please check carefully the entire manuscript.
Answer: Spelling errors in the entire manuscript have been corrected and shaded with yellow
Lines 201-206 are misunderstanding. Please rephrase.
Answer: The sentences have been restructured in line 199-203
References: the number of citing references is low. Recent papers were not cited in this review paper:
Fortification of beef burger with the addition of bee pollen from Apis mellifera L. - https://doi.org/10.9755/ejfa.2019.v31.i11.2025
The application of pollen as a functional food and feed ingredient—the present and perspectives - https://doi.org/10.3390/biom10010084
Composition and functionality of bee pollen: A review - https://doi.org/10.1016/j.tifs.2020.02.001
Answer: The references have been added in Table 2, Line bee pollen) 36-39, and 35-36, respectively.
I agree with the publishing of this paper after minor revision.

Reviewer 2 Report
Addition of Bee Products in Diverse Food Sources: Functional and Physicochemical Properties.
COMMENTS:
The review compilates some information in the scientific literature about the influence on the properties and composition of some foods when bee products are added.
It is an interesting document; however, some important changes should be done before the acceptance for publishing.
Main comments:
Firstly, a simple search using the words “Yoghurt with honey” for example gives me many other interesting references about the use of this bee product. Some are detailed below.
Varga, L. (2006). Effect of acacia (Robinia pseudo-acacia L.) honey on the characteristic microflora of yogurt during refrigerated storage. International journal of food microbiology, 108(2), 272-275.
Machado, T. A. D. G., de Oliveira, M. E. G., Campos, M. I. F., de Assis, P. O. A., de Souza, E. L., Madruga, M. S., ... & do Egypto, R. D. C. R. (2017). Impact of honey on quality characteristics of goat yogurt containing probiotic Lactobacillus acidophilus. LWT, 80, 221-229.
Coskun, F., & Karabulut Dirican, L. (2019). Effects of pine honey on the physicochemical, microbiological and sensory properties of probiotic yoghurt. Food Science and Technology, 39, 616-625.
Rotar, A. M., Semeniuc, C., Bunghez, F., Jimborean, M., & Pop, C. (2014). Effect of different storage period on lactic acid bacterias from goji yogurt and goji yogurt with honey. Bulletin UASVM Food Science and Technology, 71(1), 75-76.
My first suggestion is to improve the methodology to find information. Maybe the available scientific literature is too large, so that, to facilitate reading and understanding of the paper the second suggestion is to group the information found considering effects in foods. In this sense, it is difficult to follow what authors want to describe about the use of bee products. Many sentences are only a transcription of what other authors found. It can be seen, for example, in lines 98-107 and others. To avoid this, in my opinion, it is better to consider the effects of the addition of different bee products and then discuss what the literature said. For example, effects in the content of bifidobacteria when bee products are added to yogurt, effects in protein content when different bee products are added to foods, effect in ash content, effects in polyphenols, antioxidants or sensorial properties and so on.
Tables 1 and 2 are good, but considering the above comments, can be reorganized showing grouped the effects of adding bee products. Each of the lines can summarize the effects that have been found in different references, whether positive or negative adding at the end the corresponding references.
The column “origin” includes the country where the research was done, I suppose, but I miss the botanical origin (honey type) of honey. Different honey types have different properties and sometimes literature refers to the botanical origin. It would be very interesting if authors can add this information.
Minor comments
In some sentences, it is not clear the meaning. For example,
Lines 75-76, “However, a similar effect has also been observed in pollen and royal jelly, thus these changes are described below”, what are similar effects? These are not described before.
Lines 127-129, “Finally, the incorporation of propolis in yogurt generates similar concentrations of fatty acids, thus concluding that propolis is similar to the proportion of fatty acids compared to the control”
Lines 188-190, “Also, a mixture of yogurt with 0.05% of royal jelly does not generate harmful effects in sensory characteristics. However, it is worth mentioning that the formulations containing corn syrup and flavorings might have influenced the evaluation”
Lines 316-317, The characteristics of honey enriched with propolis at 0.1-1% intervals, did not generate significant changes in humidity, fructose, glucose, sucrose content.
Sometimes the word “and” is wrongly italicized (in table 1). The same for bifidobacteria.
Also in table 1, the column “amount added” has the units (normally %) added in each number and sometimes only at the end.
Page 4, line propolis, reference [18]. It is expected the absence of E. coli and Salmonella in yogurt, so that what are the effects of propolis?
Line 206 and 208, the number of the reference is italicized.
Line 272, change Coli for coli
Table 2 should be added before section 7.
Line 323, a proportion of >0.3%, change for a proportion higher than 3%.
Some sentences are too large and need a full stop to make comprehension easier. For example, Lines 245-249, lines 331-334.
Finally, about the conclusions, they should be according to the results. For example, In the case of honey (lines 344-346) there are no comments before about the level of acceptance that justify this conclusion. However, it seems to add honey increases the level of bifidobacteria in yogurt.
Author Response
Mexico, 24 august, 2021
We would like to thank reviewers for the comments made to our manuscript entitled “Addition of Bee Products in Diverse Food Sources: Functional and Physicochemical Properties”. We include our responses to the comments raised by the reviewers point by point, as well as, revision of all manuscript, spelling errors and the English language by a native speaker. Corrections and added information have been shaded with yellow.
REVIEWER 2
COMMENTS:
The review compilates some information in the scientific literature about the influence on the properties and composition of some foods when bee products are added.
It is an interesting document; however, some important changes should be done before the acceptance for publishing.
Main comments:
Firstly, a simple search using the words “Yoghurt with honey” for example gives me many other interesting references about the use of this bee product. Some are detailed below.
Varga, L. (2006). Effect of acacia (Robinia pseudo-acacia L.) honey on the characteristic microflora of yogurt during refrigerated storage. International journal of food microbiology, 108(2), 272-275.
Machado, T. A. D. G., de Oliveira, M. E. G., Campos, M. I. F., de Assis, P. O. A., de Souza, E. L., Madruga, M. S., ... & do Egypto, R. D. C. R. (2017). Impact of honey on quality characteristics of goat yogurt containing probiotic Lactobacillus acidophilus. LWT, 80, 221-229.
Coskun, F., & Karabulut Dirican, L. (2019). Effects of pine honey on the physicochemical, microbiological and sensory properties of probiotic yoghurt. Food Science and Technology, 39, 616-625.
Rotar, A. M., Semeniuc, C., Bunghez, F., Jimborean, M., & Pop, C. (2014). Effect of different storage period on lactic acid bacterias from goji yogurt and goji yogurt with honey. Bulletin UASVM Food Science and Technology, 71(1), 75-76.
Answer: As the reviewer suggests, most of the references were added in Table 1, however the reference Varga, L. (2006) was not added, because in this review article it is intended to provide information about most recent papers.
My first suggestion is to improve the methodology to find information. Maybe the available scientific literature is too large, so that, to facilitate reading and understanding of the paper the second suggestion is to group the information found considering effects in foods. In this sense, it is difficult to follow what authors want to describe about the use of bee products. Many sentences are only a transcription of what other authors found. It can be seen, for example, in lines 98-107 and others. To avoid this, in my opinion, it is better to consider the effects of the addition of different bee products and then discuss what the literature said. For example, effects in the content of bifidobacteria when bee products are added to yogurt, effects in protein content when different bee products are added to foods, effect in ash content, effects in polyphenols, antioxidants or sensorial properties and so on.
Answer: The methodology section has been removed at the suggestion of the first reviewer comment (reviewer comment 1 “Obviously, you searched all the international databases and for sure you did not use only those searching keywords”).
On the other hand, the sentences have been modified in order to describe in more detail the effect of adding different apiary products, line 103-104, line 119-121, line 164-165, line 188-190
Tables 1 and 2 are good, but considering the above comments, can be reorganized showing grouped the effects of adding bee products. Each of the lines can summarize the effects that have been found in different references, whether positive or negative adding at the end the corresponding references.
Answer: Tables have been reorganized
The column “origin” includes the country where the research was done, I suppose, but I miss the botanical origin (honey type) of honey. Different honey types have different properties and sometimes literature refers to the botanical origin. It would be very interesting if authors can add this information.
Answer: The column “origin” has been added
Minor comments
In some sentences, it is not clear the meaning. For example,
Lines 75-76, “However, a similar effect has also been observed in pollen and royal jelly, thus these changes are described below”, what are similar effects? These are not described before.
Answer: The sentences has been restructured in line 68-71 as follow:
“it produces changes in several characteristics (bacterial survival, nutritional composition, antioxidants content, antioxidant activity and sensory properties). Similar effect has been observed in pollen and royal………...”
Lines 127-129, “Finally, the incorporation of propolis in yogurt generates similar concentrations of fatty acids, thus concluding that propolis is similar to the proportion of fatty acids compared to the control”
Answer: The sentences has been restructured in line 121-122 as follow:
“Finally, the incorporation of propolis in yogurt generated similar fatty acids content compared to the control”
Lines 188-190, “Also, a mixture of yogurt with 0.05% of royal jelly does not generate harmful effects in sensory characteristics. However, it is worth mentioning that the formulations containing corn syrup and flavorings might have influenced the evaluation”
Answer: The sentences has been restructured in line 183-186 as follow:
“Also, a yogurt with 0.05% of royal jelly, generated pleasant sensory characteristics. “Nevertheless formulations had corn syrup and flavorings, and this could have favored the result”
Lines 316-317, The characteristics of honey enriched with propolis at 0.1-1% intervals, did not generate significant changes in humidity, fructose, glucose, sucrose content.
Answer: The sentences has been restructured in line 320-322 as follow:
“ Honey enriched with propolis at 0.1-1% intervals, no generates changes in humidity, fructose, glucose, sucrose content.”
Sometimes the word “and” is wrongly italicized (in table 1). The same for bifidobacteria.
Answer: The word “and” has been corrected
Also in table 1, the column “amount added” has the units (normally %) added in each number and sometimes only at the end.
Answer: The symbol % was only added at the end
Page 4, line propolis, reference [18]. It is expected the absence of E. coli and Salmonella in yogurt, so that what are the effects of propolis?
Answer: The information was corrected, Table 1, reference 19
Line 206 and 208, the number of the reference is italicized.
Answer: The numbers have been corrected in line 204 and 206
Line 272, change Coli for coli
Answer: Coli for coli has been corrected in Table 1, reference 19
Table 2 should be added before section 7.
Answer: Table 2 has been added before section 7
Line 323, a proportion of >0.3%, change for a proportion higher than 3%.
Answer: The sentence has been changed in line 328
Some sentences are too large and need a full stop to make comprehension easier. For example, Lines 245-249, lines 331-334.
Answer: The sentences have been restructured in line 245-253 as follow:
“A similar protective effect was found in salami samples (0.05% propolis extract) in comparison to the positive control by using BHT (butylated hydroxyltoluene) after 35 days of aging. Also, in a burger meat with microencapsulated propolis (0.1 g/kg) added, showed a better lipid oxidative stability than by using a commercial additive (sodium erythorbate)”. The burger meat-propolis added showed lower TBARS (thiobarbituric acid) value than control during storage. The maximum malonaldehyde value was reached in 14 days, while to the control sample it was reached at 7 days. This could be due to the gradual propolis bioactive compounds released. However, for this product, the sensory characteristics and the general consumer acceptance was negatively affected”
and line 335-337 as follow:
“A mixture of honey with dry cherries (40%), had better acceptability scores, even stored for 3 months, moreover the total phenolic and flavonoids contents increase up to 2.17 and 2.81 more times, respectively, this could be due to cherries are antioxidant-rich source
Finally, about the conclusions, they should be according to the results. For example, In the case of honey (lines 344-346) there are no comments before about the level of acceptance that justify this conclusion. However, it seems to add honey increases the level of bifidobacteria in yogurt.
Answer: The conclusion has been restructured by removing the word “acceptance level”.
Round 2
Reviewer 2 Report
No more comments